# Spatiotemporal Patterns of Tuberculosis in Hunan Province, China

**DOI:** 10.3390/ijerph18136778

**Published:** 2021-06-24

**Authors:** Kefyalew Addis Alene, Zuhui Xu, Liqiong Bai, Hengzhong Yi, Yunhong Tan, Darren J. Gray, Kerri Viney, Archie C. A. Clements

**Affiliations:** 1Faculty of Health Sciences, Curtin University, Perth 6102, Australia; archie.clements@curtin.edu.au; 2Wesfarmers Centre of Vaccines and Infectious Diseases, Telethon Kids Institute, Perth 6009, Australia; 3Department of Tuberculosis Control, Tuberculosis Control Institute of Hunan Province, Changsha 410000, China; xuzuhui@126.com; 4Department of Director’s Office, Hunan Chest Hospital, Changsha 410013, China; liqiong99@126.com; 5Department of MDR-TB, Internal Medicine, Hunan Chest Hospital, Changsha 410013, China; hengzhongyi@163.com (H.Y.); tanyunhong@163.com (Y.T.); 6Research School of Population Health, the Australian National University, Canberra 2601, Australia; darren.gray@anu.edu.au (D.J.G.); vineyk@who.int (K.V.); 7Department of Global Public Health, Karolinska Institutet, 141 83 Stockholm, Sweden; 8School of Public Health, The University of Sydney, Sydney 2006, Australia

**Keywords:** tuberculosis, spatial analysis, spatiotemporal, China

## Abstract

Tuberculosis (TB) is the leading cause of death from a bacterial pathogen worldwide. China has the third highest TB burden in the world, with a high reported burden in Hunan Province (amongst others). This study aimed to investigate the spatial distribution of TB and identify socioeconomic, demographic, and environmental drivers in Hunan Province, China. Numbers of reported cases of TB were obtained from the Tuberculosis Control Institute of Hunan Province, China. A wide range of covariates were collected from different sources, including from the Worldclim database, and the Hunan Bureau of Statistics. These variables were summarized at the county level and linked with TB notification data. Spatial clustering of TB was explored using Moran’s I statistic and the Getis–Ord statistic. Poisson regression models were developed with a conditional autoregressive (CAR) prior structure, and with posterior parameters estimated using a Bayesian approach with Markov chain Monte Carlo (MCMC) simulation. A total of 323,340 TB cases were reported to the Hunan TB Control Institute from 2013 to 2018. The mean age of patients was 51.7 years (SD + 17.6 years). The majority of the patients were male (72.6%, *n* = 234,682) and had pulmonary TB (97.5%, *n* = 315,350). Of 319,825 TB patients with registered treatment outcomes, 306,107 (95.7%) patients had a successful treatment outcome. The annual incidence of TB decreased over time from 85.5 per 100,000 population in 2013 to 76.9 per 100,000 population in 2018. TB case numbers have shown seasonal variation, with the highest number of cases reported during the end of spring and the beginning of summer. Spatial clustering of TB incidence was observed at the county level, with hotspot areas detected in the west part of Hunan Province. The spatial clustering of TB incidence was significantly associated with low sunshine exposure (RR: 0.86; 95% CrI: 0.74, 0.96) and a low prevalence of contraceptive use (RR: 0.88; 95% CrI: 0.79, 0.98). Substantial spatial clustering and seasonality of TB incidence were observed in Hunan Province, with spatial patterns associated with environmental and health care factors. This research suggests that interventions could be more efficiently targeted at locations and times of the year with the highest transmission risk.

## 1. Background

Tuberculosis (TB) is an ancient communicable disease caused by the bacterium *Mycobacterium tuberculosis* [1,2]. It is transmitted through the respiratory system when the organism is aerosolized by the cough of an infected patient and inhaled into the alveoli of a new host [3]. TB has killed approximately two billion people in the last two centuries [4], and it remains the leading cause of death from a bacterial pathogen worldwide, with more than 10 million people becoming newly sick from TB each year [5]. According to the World Health Organization (WHO) 2019 report, there were an estimated 10.0 million cases of TB (equivalent to an annual incidence of 140 cases per 100,000 population) and 1.4 million deaths due to TB in 2019 [6]. The highest incidence rate and about 87% of the TB cases in the world occur in the 30 high-TB burden countries [7]. The Southeast Asia region accounts for 44% of all TB cases [8].

China has the third highest TB burden in the world, with an estimated 833,000 cases and 31,000 deaths due to TB in 2019 [6]. The country is striving to reduce the magnitude of TB in line with the global End-TB Strategy [9]. Several interventions such as adoption and implementation of the End-TB Strategy, expansion of directly observed therapy (DOT), integration of TB and human immunodeficiency virus (HIV) care, and expansion of MDR-TB diagnosis and treatment centers have been implemented in China to reduce the burden of TB in the country. As a result, China has been able to reduce the incidence of TB by 3.4% per year since 1990 [10]. Despite this progress, TB is still a major public health problem in China, and the country has been classified by WHO as one of the 30 high-TB burden countries in the world [10].

Measuring the burden of TB by geographic area is important for TB control programs and health care providers to aid in planning, implementing, monitoring, and evaluating TB control efforts. Identifying areas where TB is geographically concentrated is particularly important in high-TB burden countries such as China to achieve the goals of the End-TB Strategy and to inform the rational allocation of resources for targeted interventions. Previous studies conducted in Ethiopia by our team reported spatial clustering of TB at the sub-national level and identified ecological factors associated with TB clustering [11,12]. Similarly, studies conducted in China showed spatial clustering of TB at the county and prefecture levels [13,14]. However, the studies conducted in China did not investigate drivers of the clustering, including sociodemographic and physical environmental drivers [13,14]. In 2017, Hunan Province was highly affected by TB epidemics [15]. This study therefore characterizes the spatial and temporal distribution and social and environmental drivers of TB in Hunan Province, China.

## 2. Methods

### 2.1. Study Area

This study was conducted in Hunan Province, which is located in the South Central region of China between 108°47′–114°16′ E and 24°37′–30°08′ N [16]. Hunan is one of the largest provinces in China, with a total population of approximately 72 million people and a total area of about 211,800 square kilometers [16]. Altitude ranges from 16 m in the north to 2079 m in the south. Hunan has a humid subtropical climate with four distinct seasons including short, cool, damp winters, and very hot and humid summers. The annual average rainfall is between 1200 and 1700 mm, and the annual average temperature ranges from 3 °C in January to 30 °C in July [16]. The total number of annual sunshine hours is between 1300 and 1800.

### 2.2. Data Sources

For this study, a dataset containing TB cases reported from 2013 to 2018 (six years) was obtained from the Hunan Tuberculosis Control Institute. TB is listed as a notifiable disease by the Chinese Center for Disease Control (CDC), and health professionals are responsible for the collection and entry of the data from notified patients into an internet-based TB management information system. The reporting system is managed by the provincial and national TB programs and contains demographic and clinical information such as age, sex, occupation, address, date of TB symptom onset, date of diagnosis, results of smear microscopy, type of TB, history of TB treatment, and treatment outcomes. Sensitive information such as names of patients and phone numbers were excluded in this study because of privacy and confidential issues. Diagnosis of TB is based on clinical signs, chest X-ray, and laboratory confirmation using smear microscopy and culture. This study included all forms of TB, including bacteriologically confirmed and clinically diagnosed TB, previously treated and new TB, and childhood and adult TB.

The TB data obtained from the Hunan Tuberculosis Control Institute were aggregated at the county level and linked with ecological-level data obtained from different sources. A wide range of socioeconomic, environmental, and health care access variables were collected from free online sources, including the Worldclim database, and the Atlas of Population Density [17]. The number of people living in each county was obtained from the Hunan Bureau of Statistics official estimates.

### 2.3. Data Analysis

Standardized morbidity ratios (SMRs) for each county were calculated by dividing the observed number of cases by the expected number. The expected number of cases was calculated by multiplying the provincial incidence by the average population for each county. Variance inflation factors (VIF) were used to check for the presence of multicollinearity among covariates. The proportion of medical personnel per 100,000 population had a VIF of 14 and was excluded from the subsequent models. Spatial autocorrelation was explored at a global scale using Moran’s I statistic and at a local scale using local indicators of spatial association (LISA), estimated using the Anselin Local Moran’s I statistic, and the Getis–Ord statistic. The LISA and the Getis–Ord statistics were used to detect local clustering of TB and to identify the locations of hotspots.

A generalized linear modeling approach was used to further define hotspots and quantify environmental drivers of TB risk. Bivariate analysis was conducted, and variables with a *p*-value < 0.2 were selected for a multivariate Bayesian spatial model.

Bayesian spatial Poisson regression models were constructed using the WinBUGS software, version 1.4.3 (Medical Research Council Biostatistics Unit, Cambridge, UK). The outcome was the number of TB cases in each county, and the offset was the expected number of cases. All variables selected from the bivariate analysis were incorporated as fixed effects, and spatially structured and unstructured random effects for each county were included in the model. The spatially structured random effects were computed using a conditional autoregressive (CAR) prior structure. The details of the Bayesian spatial Poisson regression models are provided as Appendix A.

The posterior parameter was estimated from the prior and data likelihood information using a Bayesian Markov chain Monte Carlo (MCMC) simulation approach. Convergence of the model was checked using visual inspection of posterior kernel densities and history plots. The best fitting and most parsimonious model was selected based on the lowest value of the deviance information criterion (DIC).

### 2.4. Ethical Clearance

Ethical clearance was secured from the Australian National University and Curtin University Human Research Ethics Committees for this study. Additional approval of the study and written permission to access the TB data were obtained from the Hunan Tuberculosis Control Institute.

## 3. Results

### 3.1. Demographic Characteristics of Tuberculosis

Table 1 shows the demographic and clinical characteristics of TB patients. There was a total of 323,340 TB cases reported to the Hunan TB Control Institute from 2013 to 2018, of which 234,682 (72.61%) were male and 251,586 (77.81%) were farmers by occupation. The mean age of the patients was 51.7 years (SD 17.6 years), with a range of 110 years (minimum less than 1 and maximum 110). Approximately one third (32.4%, 104,906) of the patients were referral cases, and a similar number (35.5%, 114,731) of patients were diagnosed when symptomatic individuals sought care at health care facilities.

### 3.2. Clinical Characteristics of Tuberculosis

Overall, 315,359 (97.53%) of the cases were diagnosed with pulmonary TB, and treatment outcomes were registered for 319,954 patients. Of those patients with registered treatment outcomes, 113,106 (35.35%) were cured and 193,125 (60.36%) successfully completed treatment without bacteriologic evidence of cure. Overall, 306,231 (95.71%) patients had successful treatment outcomes (i.e., cured or treatment completed), 6679 (2.09%) patients had treatment failure, 1020 (0.32%) were lost to follow-up, and 3188 (1.00%) died. The number of patients with treatment failure steadily increased from 89 cases in 2013 to 557 cases in 2018, and similarly, numbers switching to MDR-TB treatment increased from 63 in 2013 to 413 in 2018. By contrast, the number of patients who were lost to follow-up and TB-related deaths was relatively constant throughout the study period; 188 in 2013 to 117 in 2018 for lost to follow-up, and 144 in 2013 to 127 in 2018 for TB-related deaths.

### 3.3. Temporal Trends of Tuberculosis Incidence

The overall crude incidence of TB for the six-year period was 478.8 per 100,000 population, with prefecture-level rates ranging from 317.2 per 100,000 population in Yueyang to 640.1 per 100,000 population in Zhangjiajie. The annual incidence of TB steadily decreased over time from 85.5 per 100,000 population in 2013 to 76.9 per 100,000 population in 2018. TB cases have shown seasonal variations with the highest number of TB cases reported in late spring and early summer (March–May) in all years (Figure 1).

### 3.4. Spatial Clustering of Tuberculosis

TB incidence varied substantially at the county level, with SMRs ranging from 0.08 in Pingjiang to 7.69 in Hongjiang Shi (Figure 2). A hotspot was observed in the western part of the province (global Moran’s I = 0.25; *p* = 0.014; Figure 3). After incorporating the socio-climatic variables in the model, the posterior mean of spatially structured random effects for TB incidence showed spatial clustering of high risk in the far western and far eastern parts of Hunan Province and low risk in the area north of the capital city, Changsha (global Moran’s I = 0.15; *p* = 0.04; Figure 4).

### 3.5. Factors Associated with Spatial Clustering of Tuberculosis

Table 2 shows that in the univariate analysis, the percentage of urban residents in the counties, the GDP of the counties, and the prevalence of contraceptive use were significantly associated with the incidence of TB. In the best fitted multivariate Bayesian model (Table 3), there was a significant, negative association between both monthly sunshine exposure (relative risk (RR): 0.86; 95% credible interval (CrI): 0.74, 0.97) and contraceptive use (RR: 0.88; 95% CrI: 0.79, 0.98) with TB incidence. The model also showed that the temporal trends of TB declined over the study period (RR: 0.63; 95% CrI: 0.54, 0.82).

## 4. Discussion

This study presented a spatial analysis of TB at the county level in Hunan Province using routinely collected TB surveillance data. The main result of the study indicates geographical clustering of TB in Hunan Province, which was significantly associated with climatic and health care access factors. Similar spatial clustering of TB was reported in previous studies conducted in China [14,18,19] and other high-TB burden countries such as Peru [20], Ethiopia [21,22], and India [23]. There could be several reasons for the observed spatial clustering of TB. One reason could be the impact of the proximity and localized movement of people on the transmission of TB, causing areas neighboring high-incidence counties to be at greater risk [24]. The other reason for clustering could be that counties close to each other have similar geographic characteristics that relate to the transmission of the infection and development of the disease, which leads to a similar incidence. The high-incidence cluster in the western part of Hunan Province could be impacted as a result of poor socioeconomic conditions and poor health care access. The practical implication of this finding is that TB control in Hunan Province will need to be targeted at a sub-provincial level (such as the county) for more cost-effective use of resources.

The annual decline in TB incidence reported here appears to be due to many factors, including implementation of community-based TB activities throughout the province, or conversely due to increasing under-reporting of TB cases during the period [6]. To be confident that interventions have had a positive impact, the latter possibility needs to be explored further.

Our study showed that TB incidence in Hunan Province was seasonal. Winter conditions may cause new infections as well as exacerbation of symptoms that lead to a diagnosis. As TB has a long incubation period, ranging from months to years, winter conditions causing increased indoor activities and crowding, cold environments with low humidity, and reduced exposure to sunshine might explain the subsequent peak in TB incidence in late spring/early summer. Our model, assessing the relationship between hours of sunshine and TB incidence, found that low levels of sunshine were significantly associated with a high incidence of TB. This might be related to the potential link between sunlight and vitamin D levels. Reducing exposure to sunshine in the winter season may decrease vitamin D levels that may consequently impair host defenses to TB [25,26]. However, the patterns of TB incidence could also be influenced by other confounding factors such as behavioral, dietary, and environmental factors which were not assessed in our study.

Our study also identified some ecological-level risk factors that play an important role in the spatial clustering of TB in Hunan Province. For example, health care access indicators such as the low prevalence of contraceptive use were significantly associated with the incidence of TB. This finding is similar to our previous studies conducted in Ethiopia that showed a significant association between the incidence of TB and health care access factors [21,27,28,29]. Such findings may indicate that strengthening health care access, including the diagnosis and treatment of TB, especially in hotspot areas, may help to reduce the burdens of TB and achieve national End-TB strategies.

Although several socioeconomic and climatic variables were included in our study, some other important variables that showed a significant association with TB clustering in previous studies such as poor living conditions, nutrition, exposure to ambient air pollution, and smoking were not available to be included in our analysis [30,31]. Due to the nature of our ecological study, which used surveillance data aggregated at the county level, it was not possible to directly quantify the impact of individual-level factors affecting TB incidence. In addition, since our study was conducted only in one province in South Central China, economic, cultural, and geographic variations may limit the generalizability of the findings to other provinces in China. Finally, since our study used surveillance data collected from a passive surveillance system, under-reporting of TB cases would be an important limitation that has an impact on the identification of clusters and risk factors. However, according to the recent WHO TB report, China was one of the countries with lowest levels of under-reporting. Moreover, TB diagnosis and treatment in Hunan Province are provided free of charge, and patients receive reimbursement for transportation and food expenses, which may reduce under-detection and under-reporting of cases.

## 5. Conclusions

This study identified spatial clustering of notified TB at the county level in Hunan Province, China. Hotspots of high TB incidence were observed in counties located in the far west and far east of the province. Climatic and health care access variables such as sunshine hours and prevalence of contraceptive use were significantly associated with the clustering of TB. TB incidence in Hunan Province reveals seasonality, with a peak at the end of spring and the beginning of summer. Resources should be prioritized at a sub-provincial level when and where they are likely to be most effective.

## Figures and Tables

**Figure 1 ijerph-18-06778-f001:**
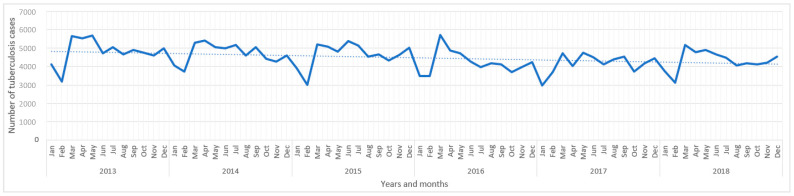
The monthly reported number of tuberculosis cases in Hunan Province, China, 2013–2018. The dotted line shows the linear trends of tuberculosis over the study period.

**Figure 2 ijerph-18-06778-f002:**
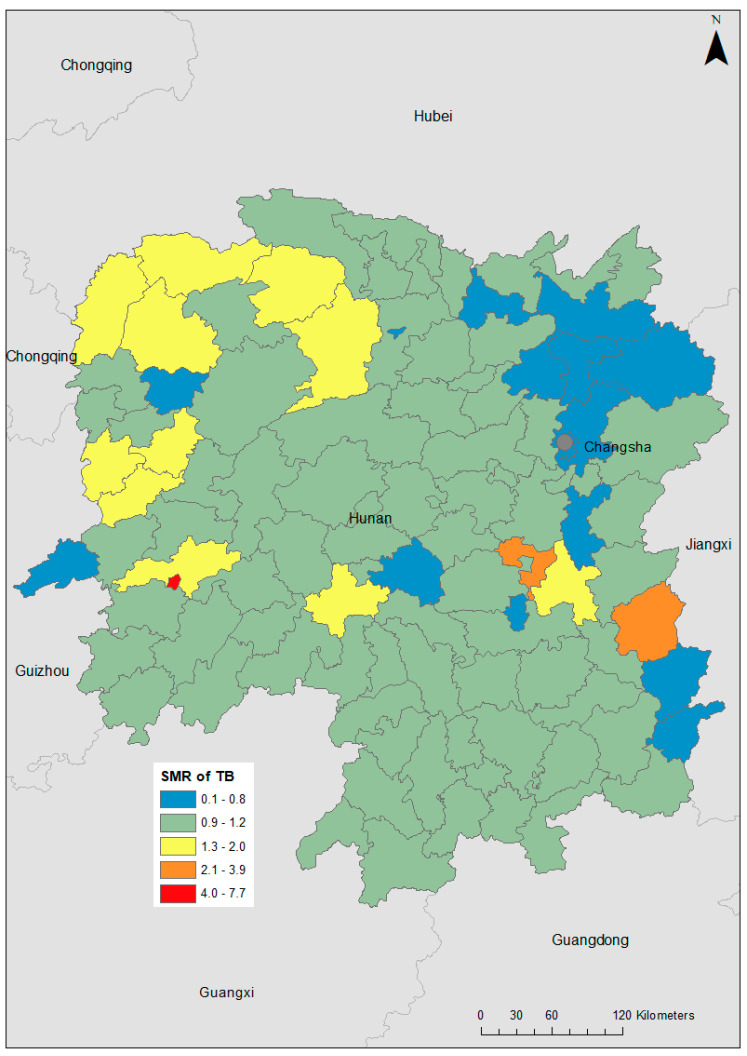
Choropleth map showing the geographical distribution of tuberculosis (TB) standardized morbidity ratios (SMR) across each county in Hunan Province, 2013–2018.

**Figure 3 ijerph-18-06778-f003:**
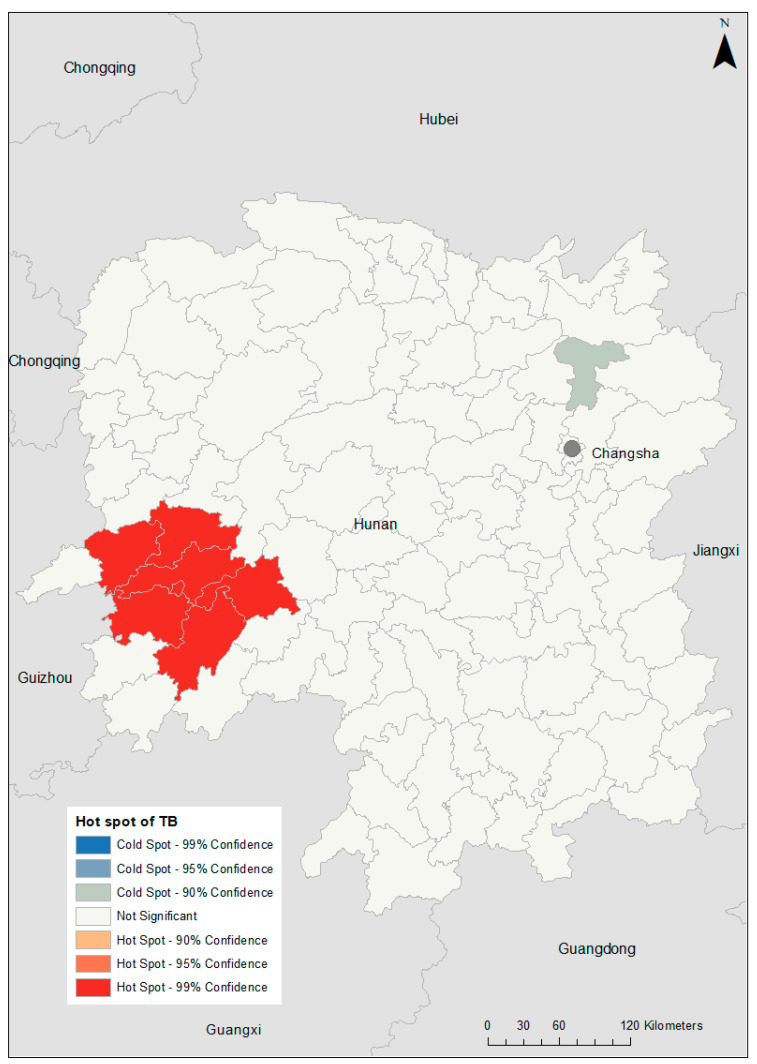
Spatial clustering of tuberculosis (TB) incidence in Hunan Province, 2013–2018, based on the Getis–Ord Gi* statistic.

**Figure 4 ijerph-18-06778-f004:**
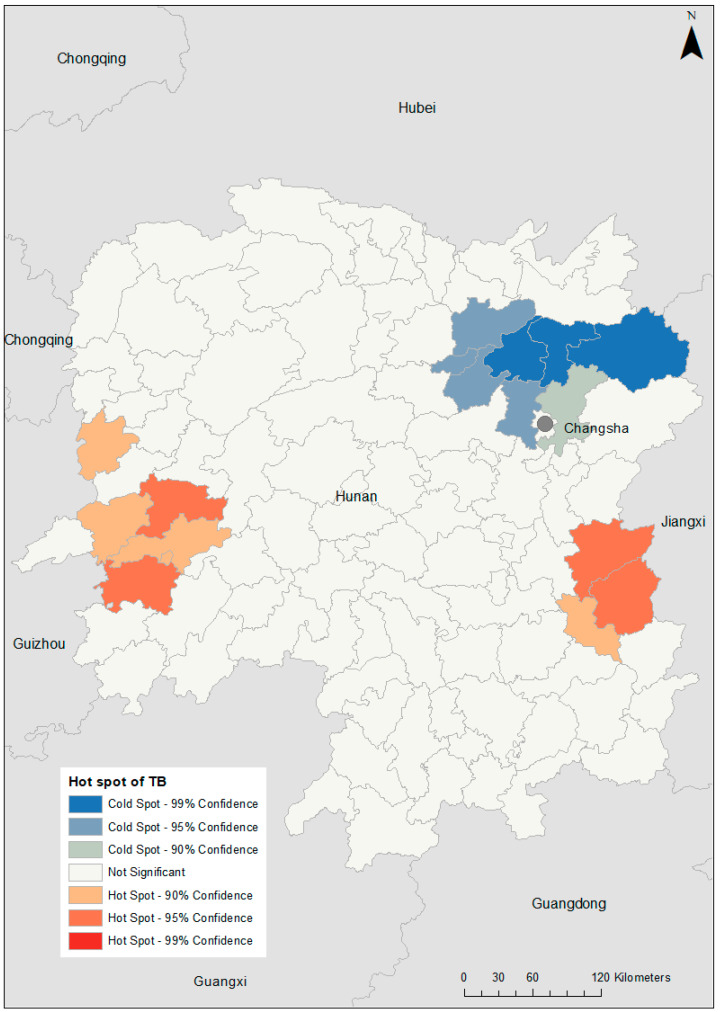
Posterior mean of spatially structured random effects for tuberculosis (TB) incidence in Hunan Province, 2013–2018.

**Table 1 ijerph-18-06778-t001:** Demographic and clinical characteristics of tuberculosis patients in Hunan Province, 2013–2018.

Variables	Number (Percent)
Mean age (standard deviation)	51.7 (17.6)
Sex	
Male	234,682 (72.61)
Female	88,524 (27.39)
Occupation	
Farmer	251,586 (77.81)
Government employee	6167 (1.91)
Non-government employee	5041 (1.56)
Laborer	10,713 (3.31)
Retired	12,953 (4.01)
Student	9421 (2.91)
Unemployed	18,553 (5.74)
Unknown	4819 (1.49)
Others	4087 (1.26)
Year	
2013	57,890 (17.9)
2014	56,720 (17.54)
2015	55,855 (17.27)
2016	50,754 (15.7)
2017	50,088 (15.49)
2018	52,033 (16.09)
Patient sources	
Contact check	384 (0.12)
Recommended due to symptoms	4641 (1.44)
Referral	104,906 (32.44)
Seek medical treatment	114,731 (35.48)
Health examination	3206 (0.99)
Track	94,111 (29.11)
Other	1361 (0.42)
Type of tuberculosis	
Pulmonary tuberculosis	315,350 (97.53)
Extra pulmonary tuberculosis	7990 (2.47)
Treatment outcome	
Treatment completed	193,125 (60.38)
Cure	113,106 (35.36)
Failure	6679 (2.09)
Lost to follow-up	1020 (0.32)
Death	3188 (1.00)
Other	2836 (0.89)

**Table 2 ijerph-18-06778-t002:** Outputs of univariate Poisson regression models of tuberculosis incidence in Hunan Province, China, 2013–2018.

Variables	Coefficient (95% CI *)	*p*-Value
Socioeconomic and demographic factors		
Proportion of males in a county	−0.12 (−0.27, 0.04)	0.13
Percentage of urban residents in the counties	0.29 (0.06, 0.52)	0.01
Gross domestic product of the county	−0.25 (−0.45, −0.06)	0.01
Birth rate in the county	−0.17 (−0.35, 0.01)	0.06
Death rate in the county	−0.12 (−0.35, 0.12)	0.34
Health care access		
Contraceptive use rate of the county	−0.33 (−0.48, −0.17)	<0.001
Number of institutions per 10,000 population in a county	0.05 (−0.16, 0.26)	0.64
Number of hospital beds per 10,000 population in a county	−0.36 (−0.81, 0.09)	0.11
Number of medical personnel per 10,000 population in county	0.26 (−0.25, 0.77)	0.32
Climatic factors		
Monthly average temperature	0.06 (−0.10, 0.22)	0.49
Annual total precipitation	−0.07 (−0.26, 0.12)	0.46
Monthly sunshine hours	−0.09(−0.10, −0.08)	<0.001

* CI: confidence interval.

**Table 3 ijerph-18-06778-t003:** Outputs of a multivariate Bayesian Poisson regression model with spatially structured and unstructured random effects for Hunan Province, China, 2013–2018.

Variables	Spatially Unstructured Model	Spatially Structured Model	Both Spatially Structured and Unstructured Model
	(RR (95% CrI)	(RR (95% CrI)	(RR (95% CrI)
Socioeconomic and demographic factors			
Proportion of males in a county	0.96 (0.87, 1.06)	0.97 (0.85, 1.09)	0.96 (0.87, 1.07)
Percentage of urban residents in the counties	1.09 (0.96, 1.23)	1.04 (0.93, 1.15)	1.07 (0.95, 1.22)
Gross domestic product of the county	0.88 (0.77, 1.00)	0.94 (0.84, 1.05)	0.91 (0.80, 1.03)
Birth rate in the county	0.94 (0.84, 1.06)	0.96 (0.83, 1.09)	0.95 (0.84, 1.08)
Health care access			
Prevalence of contraceptive use	0.87 (0.78, 0.96)	0.91 (0.81, 1.01)	0.88 (0.79, 0.98)
Number of hospital beds per 10,000 population in a county	0.98 (0.87, 1.10)	0.99 (0.90, 1.10)	0.99 (0.88, 1.10)
Climatic factors			
Monthly sunshine hours	0.86 (0.78, 0.96)	0.84 (0.70, 1.00)	0.86 (0.74, 0.97)
Temporal trend (by quarter)	0.61 (0.51, 0.72)		0.63 (0.54, 0.82)
Heterogenicity			
Variance of spatially unstructured random effect (σ^2^)	0.46 (0.40, 0.53)		0.39 (0.09, 0.51)
Variance of spatially structured random effect (σ^2^)		0.95 (0.82, 1.10)	0.36 (0.03, 0.95)
Intercept (alpha)	−0.002 (−0.09, 0.09)	−0.01 (−0.02, 0.01)	−0.003 (−0.09, 0.08)
DIC	1217.0	816.0	643.0

RR: relative risk; CrI: credible interval; DIC: deviance information criterion.

## Data Availability

The data that support the findings of this study are available from the corresponding author on reasonable request.

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
