# Peer review of "Spatiotemporal Patterns of Tuberculosis in Hunan Province, China"

_ijerph, 2021, doi:10.3390/ijerph18136778_

Round 1

Reviewer 1 Report

The study aims to analyze the prevalence of tuberculosis in Hunan Province, China. It is within the scope of the journal. Before a further assessment, I have several comments and suggestions. Please see my comments below:

Table 2 description and Table 2 values do not match: "Table 2 shows that in the univariate analysis, the percentage of urban residents in the counties, GDP of the county and the prevalence of contraceptive use were significantly associated with the incidence of TB. In the best-fitted multivariate Bayesian model, there was a significant, negative association between both monthly sunshine exposure and con-traceptive use with TB incidence."The monthly sunshine hours does not have a P value of less than 0.05.

Not verified by Table 2: "Climatic and health care assess variables such as sunshine hours and prevalence of contraceptive use were significantly associated with clustering of TB."

Figure 4 has not been referred to in the text?

Figures and Tables: Describe all of the abbreviations used in the figures/tables in the corresponding caption.

There are many abbreviations used in the manuscript, so please add a table of abbreviations.

Author Response

Response to Reviewer 1 Comments

Point 1: The study aims to analyze the prevalence of tuberculosis in Hunan Province, China. It is within the scope of the journal. Before a further assessment, I have several comments and suggestions. Please see my comments below

Response 1: We are grateful for the thoughtful comments of the reviewer. We have carefully reviewed the comments and have revised the manuscript accordingly. 

Point 2: Table 2 description and Table 2 values do not match: "Table 2 shows that in the univariate analysis, the percentage of urban residents in the counties, GDP of the county and the prevalence of contraceptive use were significantly associated with the incidence of TB. In the best-fitted multivariate Bayesian model, there was a significant, negative association between both monthly sunshine exposure and contraceptive use with TB incidence." The monthly sunshine hours does not have a P value of less than 0.05. Not verified by Table 2: "Climatic and health care assess variables such as sunshine hours and prevalence of contraceptive use were significantly associated with clustering of TB."

Response 2: We have now checked the univariate Poisson regression analysis, table 2, and the description in the text.  The necessary corrections have been made in the revised version of the manuscript (see on page 8 line 4 and Table 2).

Point 3: Figure 4 has not been referred to in the text?

Response 3: Thank you for these comments. We have corrected this in the revised version of the manumit on page 5.

Point 4: Figures and Tables: Describe all of the abbreviations used in the figures/tables in the corresponding caption.

Response 4:  We have now described all the abbreviations used in the figures and tables in the corresponding caption and footnotes. In addition, abbreviations have been defined in parentheses the first time they appear in the abstract and main text.  

Point 5: There are many abbreviations used in the manuscript, so please add a table of abbreviations.

Response 5: We have now included a list of abbreviations in the revised version of the manuscript on page 11.

Reviewer 2 Report

Alene et al present an analysis of spatiotemporal patterns of tuberculosis in Hunan Province of China. I find a dearth of spatiotemporal data analysis in the paper (I either see spatial or temporal data) for which I have some suggestions. I also have a couple of other suggestions that might be useful.

Suggestions 

Table 1: In the title, "demographic" should be Demographic, similarly "health examination should be "Health examination". Please change incase you agree.

Figure 1: I am not sure what the character in the y axis is. I am thus not able to make much sense of the table.

Further, would it be possible for the authors to include additional data showing space-time scan statistic for eg using SaTScan etc. The authors may even want to include a table of forecasted numbers of reported cases obtained from such a model.

Figure 3: I would also like to see a space time scan plotted over the Hotspot of Cold Spot regions of Hunan. The authors may also want to inlude a table of space-time clusters of cases observed in Hunan during the given study period.

Further more I would also recommend that the authors include a table/plot showing yearly distribution of Global spatial auto-correlation (Moran's I Statistic)

Based on my suggestions the authors may then want to add further points to their methods, results and discussion section.

Author Response

Response to Reviewer 2 Comments

Point 1: Alene et al present an analysis of spatiotemporal patterns of tuberculosis in Hunan Province of China. I find a dearth of spatiotemporal data analysis in the paper (I either see spatial or temporal data) for which I have some suggestions. I also have a couple of other suggestions that might be useful.

Response 1:  We thank you for the important suggestions. We addressed all the comments raised by the reviewer and revised the manuscript accordingly.  

Point 2: Table 1: In the title, "demographic" should be Demographic, similarly "health examination should be "Health examination". Please change in case you agree.

Response 1:  It is now corrected in the revised version of the manuscript (Table 1).

Point 3: Figure 1: I am not sure what the character in the y axis is. I am thus not able to make much sense of the table.

Response 1:  We have now replaced the abbreviation “TB” with “tuberculosis” in the y axis and caption to make it clear for the readers. The y-axis shows the number of tuberculosis cases reported in each month and year.

Point 4: Further, would it be possible for the authors to include additional data showing space-time scan statistic for eg using SaTScan etc. The authors may even want to include a table of forecasted numbers of reported cases obtained from such a model.

Response 1:  Forecasting was not the focus of this study, but we have now included a space-time component in our model (Table 3). Instead of SaTScan, we have used WinBUGS as it is more than capable of implementing the analytical requirements of this study.

Point 5: Figure 3: I would also like to see a space time scan plotted over the Hotspot of Cold Spot regions of Hunan. The authors may also want to inlude a table of space-time clusters of cases observed in Hunan during the given study period.

Response 1:  Based on the reviewer comment, we have now included temporal trends in the multivariate Bayesian regression model. The model showed that the temporal trends of TB declined over the study period (RR: 0.63; 95%CrI: 0.54, 0.82).

Point 6: Furthermore, I would also recommend that the authors include a table/plot showing yearly distribution of Global spatial auto-correlation (Moran's I Statistic)

Response 1:  We have now reported the overall Global Moran’s autocorrelation statistics in the revised version of the manuscript.

Point 7: Based on my suggestions the authors may then want to add further points to their methods, results and discussion section.

Response 1: We have now thoroughly revised the methods, results, and discussion sections of the manuscript.    

Reviewer 3 Report

The manuscript submited by Alene et al., presents new information about tuberculosis (TB) distribution in Hunan province in China. They analysed carefully the spatial distribution but also they studied socio-economic, demographic and seasonality influence on TB incidence. In summary they presented a very interesting results that should be published after correction of some minor points.

Along the text, authors classified TB as the leading cause of death caused by an infectious disease, however, TB is the main cause of death due to bacterial pathogen, because some virus cause more deaths than TB.

Results have some mistakes that authors should correct: numbers of patients according to their occupation in Table 1 should be presented on the same format than others (numbers with comma). Authors should explain better the difference between treatment completed and cure patients, because some discrepancies are presented between the text and table 1. In general data presented in part 3.2. of results and table 1 should be reviewed by authors.

Summation of patients for treatment outcome on table 1 is more than 319,825 presented in results and less than the total patients with TB, so authors should revise data or explain these results.

Months in table 1 are represented by a Chinese letter, authors should use an international sign.

Table 3 and figure 4 are not referenced on the text.

In conclusion, “Reducing exposure to sunshine in the winter season may decrease vitamin D levels that may consequently impair host defences to TB” should be supported by some references because the relation between vitamin D and resistance to some infectious disease was clearly demonstrated previously.

Author Response

Response to Reviewer 3 Comments

Point 1: The manuscript submitted by Alene et al., presents new information about tuberculosis (TB) distribution in Hunan province in China. They analysed carefully the spatial distribution but also they studied socio-economic, demographic and seasonality influence on TB incidence. In summary they presented a very interesting results that should be published after correction of some minor points.

Response 1: we thank you the reviewer for the thoughtful comments.

Point 2: Along the text, authors classified TB as the leading cause of death caused by an infectious disease, however, TB is the main cause of death due to bacterial pathogen, because some viruses cause more deaths than TB.

Response 2: we have corrected the sentence as suggested in the abstract and introduction sections.

Point 3: Results have some mistakes that authors should correct: numbers of patients according to their occupation in Table 1 should be presented on the same format than others (numbers with comma). Authors should explain better the difference between treatment completed and cure patients, because some discrepancies are presented between the text and table 1. In general data presented in part 3.2. of results and table 1 should be reviewed by authors.

Response 3: We have now corrected the results both in the text and Table 1.  Definition for successful treatment outcomes is also now included in the text on page 4.

Point 4: Summation of patients for treatment outcome on table 1 is more than 319,825 presented in results and less than the total patients with TB, so authors should revise data or explain these results.

Response 4: This error is now corrected in the revised version of the manuscript.

Point 5: Months in table 1 are represented by a Chinese letter, authors should use an international sign.

Response 5: This is now corrected.

Point 6: Table 3 and figure 4 are not referenced on the text.

Response 6: They are now referenced in the text on page 5 and 8.

Point 7: In conclusion, “Reducing exposure to sunshine in the winter season may decrease vitamin D levels that may consequently impair host defences to TB” should be supported by some references because the relation between vitamin D and resistance to some infectious disease was clearly demonstrated previously.

Response 7: We have now included references to support the statement on page 9.

Round 2

Reviewer 1 Report

The manuscript has been improved substantially. I just have a few minor comments. Please see my comments below:

Figure 1 has Chinese characters.
Figure 1: Describe the dotted line.

Author Response

Response to Reviewer Comments

Point 1: The manuscript has been improved substantially. I just have a few minor comments. Please see my comments below:

Response 1: We are grateful for the thoughtful comments of the reviewer. We have carefully reviewed the comments and have revised the manuscript accordingly. 

Point 1: Figure 1 has Chinese characters.

Response 1: It is now corrected.

Point 1: Figure 1: Describe the dotted line.

Response 1: The dotted line shows the linear trends of tuberculosis cases over the study period. This information is now included in the revised version of the manuscript.